# Unravelling the Mechanisms of Heavy Metal Tolerance: Enhancement in Hydrophilic Antioxidants and Major Antioxidant Enzymes Is Not Crucial for Long-Term Adaptation to Copper in *Chlamydomonas reinhardtii*

**DOI:** 10.3390/plants13070999

**Published:** 2024-03-30

**Authors:** Julia Dziuba, Beatrycze Nowicka

**Affiliations:** Department of Plant Physiology and Biochemistry, Faculty of Biochemistry, Biophysics and Biotechnology, Jagiellonian University, Gronostajowa 7, 30-387 Kraków, Poland; julia.dziuba@student.uj.edu.pl

**Keywords:** adaptation, antioxidant enzymes, cross-tolerance, low-molecular-weight antioxidants, oxidative stress markers, redox-active metals

## Abstract

Understanding of the mechanisms of heavy metal tolerance in algae is important for obtaining strains that can be applied in wastewater treatment. Cu is a redox-active metal directly inducing oxidative stress in exposed cells. The Cu-tolerant *Chlamydomonas reinhardtii* strain Cu2, obtained via long-term adaptation, displayed increased guaiacol peroxidase activity and contained more lipophilic antioxidants, i.e., α-tocopherol and plastoquinol, than did non-tolerant strain N1. In the present article, we measured oxidative stress markers; the content of ascorbate, soluble thiols, and proline; and the activity of superoxide dismutase (SOD), catalase (CAT), and ascorbate peroxidase (APX) in N1 and Cu2 strains grown in the absence or presence of excessive Cu. The Cu2 strain displayed less pronounced lipid peroxidation and increased APX activity compared to N1. The amount of antioxidants was similar in both strains, while SOD and CAT activity was lower in the Cu2 strain. Exposure to excessive Cu led to a similar increase in proline content in both strains and a decrease in ascorbate and thiols, which was more pronounced in the N1 strain. The Cu2 strain was less tolerant to another redox-active heavy metal, namely chromium. Apparently other mechanisms, probably connected to Cu transport, partitioning, and chelation, are more important for Cu tolerance in Cu2 strain.

## 1. Introduction

“Heavy metals” is a collective term often used for metals and metalloids with a density exceeding 5 g/cm^3^. Although some of these elements are micronutrients, they are also harmful environmental pollutants. In the last centuries, human activity led to the contamination of lands and waters with heavy metals [1]. Various methods that allow us to cope with this threat include phytoremediation and wastewater treatment using microorganisms such as unicellular algae [2,3,4]. Research focused on understanding of the mechanisms of heavy metal tolerance is therefore important for obtaining strains that can survive in contaminated water. B. Pluciński and co-workers conducted research on microevolutionary processes that lead to the acquisition of copper tolerance in the model organism green microalga *Chlamydomonas reinhardtii*. They obtained strains that could grow in the presence of increased Cu concentrations and began their characterisation [5,6,7].

Copper is an essential micronutrient. Due to its redox properties, Cu is a prosthetic group in key proteins involved in electron transfer in respiration and photosynthesis, i.e., plastocyanin and cytochrome oxidase. However, photosynthetic organisms are vulnerable to supra-optimal concentrations of Cu ions. Due to its high mobility in water, Cu is considered one of the most toxic heavy metals in aquatic ecosystems [8]. Cu exerts pleiotropic negative effects on algae. It is known to disturb photosynthetic electron transfer, especially the functioning of photosystem II (PS II), by interactions with Tyr_Z_ and Tyr_D_, nonheme Fe, cyt *b*_559_, as well as sites near Q_A_, Q_B_ binding pockets, and pheophytin [9,10,11]. The substitution of Mg^2+^ in chlorophyll (Chl) by Cu^2+^ leads to a loss of excitation energy [12]. This element also exhibits inhibitory action in the dark phase of photosynthesis [11]. Cu belongs to the so-called redox-active heavy metals, which means that its ions easily undergo redox reactions in living cells. This feature is both beneficial, as it enables Cu to be an important prosthetic group in enzymes, and harmful, when such redox reactions are unwanted. The uncontrolled redox cycling of Cu ions in cells leads to the formation of reactive oxygen species (ROS), including the most reactive form, hydroxyl radical [8,13]. Furthermore, disturbance of metabolism by the toxic action of Cu may lead to secondary ROS generation [14]. The occurrence of oxidative stress was observed in green microalgae exposed to toxic concentrations of Cu ions [13,15,16,17,18,19,20,21,22,23,24,25,26,27,28,29,30,31,32,33,34].

For their protection, living organisms have evolved various ROS-detoxifying systems based on antioxidant enzymes and low-molecular-weight antioxidants [14]. Important antioxidant enzymes are superoxide dismutases (SOD) that detoxify O_2_^−•^, catalases (CAT) that detoxify H_2_O_2_, and various peroxidases that detoxify H_2_O_2_ and organic hydroperoxides. Among peroxidases, ascorbate peroxidase (APX) is very important for antioxidant protection and ROS signalling in photosynthetic cells [8]. Ascorbate (Asc) and glutathione (GSH) are important hydrophilic low-molecular-weight antioxidants of photosynthetic cells. The other compounds, such as phenolic metabolites and proline (Pro), also display antioxidant properties. Pro was also proposed to function as a metal chelator [8]. Hydrophobic low-molecular-weight antioxidants are very important for the protection of lipid membranes. They comprise carotenoids, tocopherols, and isoprenoid quinols such as plastoquinol [8].

In the former research on Cu-tolerant strains of *C. reinhardtii* that were obtained in our department by B. Pluciński, it was observed that the Cu2 strain, adapted to growth in the presence of 5.25 μM CuSO_4_, contained about 2.6 times more α-tocopherol and plastoquinol than non-tolerant strains, namely N1 and cell wall containing strain 11-32b. The Cu2 strain also displayed about 1.75 times higher activity of guaiacol peroxidase when compared to N1 strain [7]. In the present article, we extend the analysis of oxidative stress and antioxidant defence in Cu2 strain compared to N1 strain by measuring O_2_^−•^ formation; lipid peroxidation; the content of Asc, soluble thiols, and Pro; as well as the activity of APX, CAT, and SOD in these strains grown in presence of basal (0.25 μM) and increased (5.25 μM) concentration of CuSO_4_ in the medium. We also checked the tolerance of N1 and Cu2 strains to chromium, another redox-active heavy metal, for which inducing oxidative stress is an important mode of toxicity.

## 2. Results

### 2.1. The Analysis of Oxidative Stress Markers, Photosynthetic Pigments Ratios, and Antioxidant Response in Cu-Tolerant and Non-Adapted Strain

Semiquantitative analysis of O_2_^−•^ formation showed that it was approximately 40% higher in Cu-tolerant strain Cu2 when compared to non-adapted strain N1, regardless of the medium in which the algae were grown. Exposure of both strains to increased Cu concentration caused a small but statistically significant decrease in O_2_^−•^ formation (Figure 1a).

Lipid peroxidation in algae was measured by determination of thiobarbituric acid-reactive substances (TBARS). The level of TBARS was decreased in the Cu2 strain when compared to the N1 strain, regardless of Cu concentration in the medium. There were no statistically significant differences in TBARS content between algae grown in the medium with basal Cu concentration and those cultivated at the increased Cu concentration (Figure 1b).

The ratios of photosynthetic pigments are shown in Table 1. The Chl *a*/Chl *b* ratio was increased in the Cu2 strain when compared to the N1 strain, regardless of the medium used. Exposure to increased concentration of Cu did not cause a change in Chl *a*/Chl *b* ratio in tested strains. The ratio of total carotenoids to total chlorophyll was similar in all series tested.

The content of important hydrophilic low-molecular-weight antioxidants and the activity of major antioxidant enzymes in *C. reinhardtii* N1 and Cu2 strains grown either in medium containing basal Cu concentration or in medium with an increased Cu content are shown in Figure 2. The oxidation product of Asc, namely dehydroascorbate (DHA), was also measured, but its amount was small, so signal to noise ratio was too low; therefore, the results are not shown.

The content of Asc, thiols (in their reduced form), and Pro was similar in the N1 and Cu2 strains grown in the medium containing basal Cu concentration. Exposure of algae to increased Cu concentration led to a decrease in Asc and thiols content and an increase in Pro content (Figure 2a–c). There were no statistically significant differences between Asc content in N1 and Cu2 strains exposed to increased Cu concentration; however, it could be seen that the decrease in Asc content in response to increased Cu concentration was less pronounced in the Cu2 strain than in the N1 strain (Figure 2a). A similar type of response was observed for total soluble thiols. This time, the effect was much more pronounced, as the exposure of the N1 strain to increased Cu caused a nearly 90% decrease in thiols content, whereas in the Cu2 strain, this decrease was only about 50% (Figure 2b). The increase in Pro content in the presence of an increased Cu concentration was similar in both strains (Figure 2c).

When cultures were grown in the medium containing basal Cu concentration, the activity of SOD and CAT in the Cu2 strain was 2-fold lower, while the activity of APX was approximately 1.13-fold higher when compared to the N1 strain (Figure 2d–f). Exposure to increased Cu concentration led to a statistically significant decrease in SOD activity in the N1 strain, whereas for the Cu2 strain, the decrease was much less pronounced and statistically insignificant (Figure 2d). On the contrary, in the case of CAT, the exposure to increased Cu concentration caused a decrease in the activity of this enzyme in the Cu2 strain, while in the N1 strain, it remained unchanged (Figure 2e). The exposure to 5.25 μM CuSO_4_ did not cause change in APX activity in both strains (Figure 2f).

The correlation analysis of the data presented in Table 1 and Figure 1 and Figure 2 was carried out, and the results of this analysis are presented in Table 2. There was a statistically significant negative correlation between the presence or absence of adaptation to Cu and the content of TBARS and between adaptation to Cu and the activity of CAT and SOD. A positive correlation was observed for the adaptation to Cu and the normalised carotenoid content. There was a strong negative correlation between CuSO_4_ concentration in the medium and the content of Asc and thiols, whereas a positive correlation was observed between CuSO_4_ concentration and Pro content. The TBARS content was positively correlated with the activity of CAT and SOD and negatively correlated with the carotenoid content. SOD activity was positively correlated with CAT activity. The content of Asc was positively correlated with the content of soluble non-protein thiols, and both thiols and Asc were negatively correlated with the Pro content (Table 2).

### 2.2. Tolerance to Chromium in Cu-Tolerant and Non-Adapted Strain

The tolerance of the N1 and Cu2 strains to Cr toxicity was assessed using the methodology applied in [5,6], i.e., by monitoring the maximum chlorophyll fluorescence of algae grown in multi-well plates. The results of the experiment and the statistical analysis are shown in Figure 3 and Table 3. We found that variation in algae growth was shaped by the interactions of strain, Cr ions concentration, and time (Table 3). Contrary to our expectations, the Cu2 strain showed lower tolerance to Cr than the N1 strain (Figure 3). In the case of the N1 strain, the negative impact of Cr on algal growth could be seen for 120 μM K_2_Cr_2_O_7_ and higher concentrations applied, while for the Cu2 strain, the negative impact was already visible at 80 μM K_2_Cr_2_O_7_. Furthermore, in the concentration range for which growth disturbance was observed, and for the same K_2_Cr_2_O_7_ concentrations applied, growth inhibition was more pronounced in the Cu2 strain than in the N1 strain (Figure 3).

## 3. Discussion

It is widely known that oxidative stress occurs in algae exposed to toxic concentrations of heavy metal ions and that low-molecular-weight antioxidants and antioxidant enzymes play a role in response to heavy-metal-induced stress. However, the character of this response depends on many factors, such as the species, strain, heavy metal applied, dose, time of exposure, stage of growth in which algae were exposed to stress, as well as growth conditions (light intensity, photoperiod, and medium type). An enhancement of antioxidant defence is usually observed in conditions causing weak or moderate stress, whereas harsh stress causes the depletion of antioxidants and a decrease in the activity of antioxidant enzymes [8].

In the previous experiment with *C. reinhardtii* strain 11-32b containing the cell wall [33], we cultivated algae in the presence of 20 or 25 μM CuSO_4_ and measured antioxidant response after 7, 10, and 14 days. Pronounced oxidative stress was observed only after 7 days of growth. At this time, the content of total ascorbate and the activity of APX were increased when compared to control, while there were no changes in total thiols and Pro content. SOD and CAT activities were enhanced also but only for treatment with 20 μM CuSO_4_. For longer exposure times, TBARS levels and the redox state of the plastoquinone pool were similar in the Cu-treated series and in the control, while the contents of low-molecular-weight antioxidants and enzyme activities were similar or lower in stressed algae compared to the control [33]. This is why we chose 1 week of growth in the presence of Cu for the present experiment. Luis et al. [15] exposed *C. reinhardtii* to 10–200 μM CuSO_4_ for 2 days and observed a decrease in CAT activity for 150 and 200 μM CuSO_4_ and no change in SOD activity. Zheng et al. [16] exposed *C. reinhardtii* to 5 μM CuSO_4_ for 6 h and observed an increase in SOD and CAT and no changes in APX activity. Jiang et al. [17] exposed *C. reinhardtii* to 50, 100, and 250 μM CuSO_4_ and collected samples after 1, 3, and 5 days. For 100 and 250 μM CuSO_4_, they observed an increase in SOD activity and GSH content, the latter after only 1 and 5 days of exposure.

The response of other species of green microalgae to Cu was also studied. Wu et al. [35] exposed *Chlorella* sp. to 5–160 μM CuSO_4_ for 24 h and observed an increase in the Pro content. Hamed et al. [18] measured the antioxidant response in *Chlorella sorokiniana* and *Scenedesmus acuminatus* treated with 25 or 50 μM CuCl_2_ for 7 days. An increase in Asc, GSH, and Pro content as well as an increase in APX and SOD activities was reported for both species. León-Vaz et al. [36] exposed *C. sorokiniana* to 500 μM CuCl_2_ for 42 h and observed an increase in CAT but a decrease in APX activity. Mallick [19] treated *Chlorella vulgaris* with 0.25–3 mg/dm^3^ CuCl_2_ for 72 h. He reported a decrease in the GSH content for all Cu concentrations tested except the lowest one. For CuCl_2_ concentrations higher than 1 or 2 mg/dm^3^, he observed an increase in Pro content and SOD activity and a decrease in Asc content and APX and CAT activity. Qian et al. [21] observed an increase in SOD activity but no change in CAT activity in *C. vulgaris* exposed to 1.5 μM CuSO_4_ for 48 h. Chen et al. [23] exposed *C. vulgaris* to 1–5 μM CuCl_2_ for 72 h and reported an increase in SOD activity. In their experiment, CAT activity increased for lower and decreased for higher applied Cu concentrations. Lu et al. [34] treated *Chlorella pyrenoidosa* with 0.18–10.8 mg/dm^3^ CuSO_4_ for 96 h and reported an increase in SOD activity only for 5.8 and 10.8 mg/dm^3^ CuSO_4_. Sabatini et al. [26] exposed *Scenedesmus vacuolatus* to 6.2, 108, 210, and 414 μM CuCl_2_ for 7 days and observed an increase in GSH content and SOD and CAT activities for the highest concentrations tested. A decrease in soluble non-protein thiols was observed in *Scenedesmus bijugatus* exposed to 50, 100, and 200 μM CuSO_4_ for 3 days [25]. On the other hand, an increase in soluble thiols was observed in *Selenastrum capricornutum* (*Raphidocelis subcapitata*) exposed to 0.149–1.491 mg/dm^3^ CuCl_2_ for 96 h [31]. The GSH content was increased in *S. capricornutum* exposed to 1.3 μM Cu(NO_3_)_2_ for 72 h when compared to the control, while no increase was observed for the lower Cu concentrations tested [30]. Exposure of *Scenedesmus quadricauda* to 150 μM CuCl_2_ for 24 h led to an increase in APX activity [27]. An increase in APX activity was also observed in *Dunaliella salina* and *Dunaliella tertiolecta* after 24 h of exposure to 1–20 μM CuCl_2_ [32] and in *S. capricornutum* exposed to 5–100 μM CuSO_4_ for 32 h [37]. Dewez et al. [38] treated *Scenedesmus obliquus* with 1, 2, and 3 mg/dm^3^ CuSO_4_ and collected samples after 12, 24, and 48 h. They observed an increased activity of APX and CAT. Tripathi et al. [29] exposed *Scenedesmus* sp. to 2.5 and 10 μM CuCl_2_ for 7 days and observed an increase in SOD activity and Pro content. CAT activity was increased only for 2.5 μM CuCl_2_, whereas APX was increased for 2.5 μM and decreased for 10 μM CuCl_2_.

A summary of the present experiment is shown in Figure 4.

The literature data mentioned above combined with the fact that the Cu2 strain showed an increased guaiacol peroxidase activity and an increased content of α-tocopherol and plastoquinol [7] made us expect that the other low-molecular-weight antioxidants and antioxidant enzymes would also be enhanced compared to the N1 strain. But this was not the case, with the exception of slightly increased APX activity (Table 1, Figure 2).

Taking into account the markers of oxidative stress, surprisingly, the O_2_^−•^ formation was increased in the Cu2 strain when compared to the N1 strain (Figure 1). This may be correlated with the decreased SOD activity in this strain (Figure 2d). The decreased lipid peroxidation in the Cu2 strain when compared to the N1 strain may be due to the presence of an increased amount of plastidic lipophilic antioxidants in the Cu-tolerant strain [7]. Both α-tocopherol and plastoquinol are known to be very effective scavengers of O_2_^−•^ and lipid radicals. These compounds are able to prevent initiation and inhibit propagation of lipid peroxidation [8]. Exposure to 5.25 μM CuSO_4_ did not cause pronounced oxidative stress in any of the strains. Perhaps this was a reason why no increase in antioxidant enzyme activities was observed in both strains when grown in the medium containing 5.25 μM CuSO_4_.

The Pro content was not constitutively increased in the Cu2 strain when compared to the N1 strain, but the exposure of both the N1 and Cu2 strains to 5.25 μM CuSO_4_ led to an increase in the level of Pro. As could be seen in above-mentioned literature, an increase in Pro usually occurs in green microalgae exposed to toxic concentrations of Cu. The content of Asc and soluble thiols decreased in both strains tested in response to 5.25 μM CuSO_4_, most probably due to oxidation and, in the case of thiols, also as a result of the chelation of Cu ions. This decrease was less pronounced in the Cu2 strain, allowing us to conclude that the degradation/complexation of the above-mentioned compounds occurred in the Cu-treated Cu2 strain to a lesser extent when compared to the stressed N1 strain. The correlation between Asc and thiols is understandable because GSH, which is usually the most abundant non-protein thiol in cells, plays an important role in Asc recycling [8].

It should be noted here that the relationship between enhanced antioxidant defence and heavy metal tolerance can vary. Sabatini et al. compared the tolerance to Cu and the antioxidant response of *Chlorella kessleri* and *S. vacuolatus*. *S. vacuolatus* was more tolerant to Cu and showed a more pronounced antioxidant response [26]. On the other hand, when Hamed et al. compared the tolerance to Cu and the antioxidant response in *S. acuminatus* and *C. sorokiniana*, it was found that the antioxidant response in more tolerant *S. acuminatus* was either similar or less pronounced than in *C. sorokiniana* [18]. Interestingly, in the first case, the more tolerant species accumulated more Cu than the less tolerant one, while in the second case, increased tolerance was accompanied by a decrease in accumulation of Cu [18,26]. It can be concluded that various mechanisms of tolerance to Cu interact with each other. The results of our research on the Cu-exposed *C. reinhardtii* strain 11-32b containing cell wall showed that oxidative stress and the antioxidant response were pronounced during the first 7 days of cultivation. A reason for such an effect could be the fast growth of the culture during the early phase as well as the progress of complexation and/or precipitation of Cu ions over time. However, it is also possible that longer exposure to Cu ions led to the development of protective mechanisms enabling to decrease the concentration of free Cu ions inside the cell [33]. The results of the present experiment let us suggest that other, more ion-specific mechanisms are the reason for increased tolerance to Cu in the Cu2 strain.

A wide array of protective mechanisms unrelated to antioxidant defence have been described in living organisms including higher plants, algae, and fungi [8,39,40]. Among them are those aimed at preventing the entry of metal ions into the cell. Heavy metal ions can be bound to the cell wall and excreted mucilaginous material, which is usually polysaccharide. Algae and plant roots can also release metal-binding exudates such as various polyphenols and organic acids [8,39]. Some organisms carry out bioconversion of metal ions, leading to the formation of insoluble oxides and salts [8]. An important strategy is the modulation of ion transport, i.e., active efflux, reduction of uptake, and, in higher plants, limitation of heavy metal ions translocation from the roots to the shoot [39,40]. Another protective mechanism is based on the chelation and sequestration of heavy metal ions inside the cell. Key chelating compounds are phytochelatins: short-chain, thiol-containing peptides produced enzymatically. Binding of Cu ions by phytochelatins has been observed in higher plants [41]. Cells also produce small cytosolic proteins capable of binding metal ions with high affinity, which are called metallothioneins [8]. Heavy metal ions, free or chelated, are exported to vacuoles, where they are sequestered. In vacuoles, metal ions can be bound to phytochelatins, organic acids, and polyphosphates [8,39,42]. All protective mechanisms present in certain organism cooperate to counteract heavy metal toxicity.

The role of more ion-specific protective mechanisms in Cu2 strain is supported by the results concerning tolerance of the N1 and Cu2 strains to another redox-active metal, chromium (Figure 3). This metal was applied as K_2_Cr_2_O_7_, which means that Cr was in an anionic form contrary to Cu. The increase in α-tocopherol and plastoquinol content was observed in Cr-exposed *C. reinhardtii* [43,44], while guaiacol peroxidase activity was increased in Cr-treated *C. vulgaris* [45]. However, the increased content of the above-mentioned antioxidants and activity of guaiacol peroxidase in the Cu2 strain [7] was not sufficient to provide tolerance to Cr. In fact, the Cu2 strain turned out to be less tolerant to Cr than the N1 strain. Chromate ions are taken into cells by the sulphate uptake system [46]. It may be that the Cu2 strain has enhanced sulphate uptake for the sake of synthesis of phytochelatins and/or metallothioneins. This would be beneficial for Cu tolerance but would result in enhanced Cr uptake into the cells. Furthermore, the Cu2 strain did not show enhanced tolerance to the redox-inactive heavy metals cadmium and zinc [5]. Mechanisms of Cu tolerance in the Cu2 strain require further study, which is planned in the future.

## 4. Materials and Methods

### 4.1. Algal Strains

The parent strain used in the research on long-term adaptation to Cu was a cell-wall-deficient mutant (CW15) of *Chlamydomonas reinhardtii* obtained from Dr. Itzhak Ohad, Hebrew University (Department of Biological Chemistry, Givet Ram, Jerusalem, Israel) [5]. This strain was cultured in modified Sager-Granick medium supplemented with an osmoprotectant and the source of organic carbon (see below), in Erlenmeyer flasks (250 mL), on the shaker, at 22 °C, under continuous white light (50 μmol photons m^−2^ s^−1^). Cu-tolerant strains were obtained by B. Pluciński by long-term adaptation to increased Cu concentration, as described in [5,6]. For the present experiment, two strains were chosen: the non-adapted strain N1 and the strain Cu2 continuously grown at the CuSO_4_ concentration 5.25 μM. The Cu2 strain was the first Cu-adapted strain obtained by B. Pluciński. The CuSO_4_ concentration 5.25 μM was chosen because it caused significant growth reduction in the parent strain, but it was still not very high. The long-term adapted strain was meant to be used for further studies also related to ecophysiology, while it is known that very high Cu concentrations are found in the natural environment only rarely and in extremely polluted areas [33]. During further research [5,6,7], the results obtained for the Cu2 strain were found to be very interesting, especially the increase in lipophilic antioxidants and guaiacol peroxidase activity, when compared to the parent strain [7]. Therefore, we decided to continue our studies on this strain.

### 4.2. Growth Conditions

The algae were grown in Erlenmeyer flasks (250 mL), on a shaker, at 21 °C, under continuous white light (35–40 μmol photons m^−2^ s^−1^) in the mannitol- and organic carbon-supplemented modified SG medium comprised of 16.2 μM H_3_BO_4_, 3.5 μM ZnSO_4_·7H_2_O, 2.02 μM MnCl_2_·4H_2_O, 0.84 μM CoCl_2_·6H_2_O, 0.25 μM CuSO_4_·5H_2_O, 0.83 μM (NH_4_)_6_Mo_7_O_24_·4H_2_O, 3.75 mM NH_4_NO_3_, 0.57 mM K_2_HPO_4_, 0.73 mM KH_2_PO_4_, 1.22 mM MgSO_4_·7H_2_O, 0.36 mM CaCl_2_·2H_2_O, 0.37 mM FeCl_3_, 7.5 mM sodium acetate, 1.7 mM sodium citrate, and 100 mM mannitol. They were regularly inoculated into fresh medium for at least two months prior to the experiments. The N1 strain was grown in modified SG medium with a basal Cu concentration, while the Cu2 strain was grown in the medium with 5.25 μM CuSO_4_. For the experiment, each strain was inoculated to both basal medium and Cu-enriched medium. Basal medium was used to reflect the non-stressful conditions, while 5.25 μM CuSO_4_ was used because this was the copper concentration to which the Cu2 strain was adapted, which was stressful for the parent non-tolerant strain N1. Then, 100 mL of liquid medium in an Erlenmeyer flask was inoculated with the volume of grown culture of *C. reinhardtii*, providing an initial Chl *a* + *b* concentration of 0.5 μg/mL. The algae were then grown in the conditions described above for 1 week, and the samples were collected from four separate cultures per each series.

### 4.3. Sample Collection, Measurement of Photosynthetic Pigments, and Cells Lysis Protocol

For the measurements of photosynthetic pigments, 2 mL of each culture was taken. The pigments were extracted with acetone as described in [44], and their content was determined spectrophotometrically according to method described by Lichtenthaler [47].

For the measurements of TBARS, low-molecular-weight antioxidants, and antioxidant enzymes, 30 mL of culture was taken per sample. Cell suspensions were centrifuged (8 min, 5500× *g*, 4 °C), re-suspended in 1.5 mL of fresh growth medium with basal Cu concentration, and centrifuged again (4 min, 9000× *g*, 4 °C). In the case of the determination of TBARS, low-molecular-weight antioxidants, as well as CAT and SOD activity, the pellets were frozen in liquid N_2_ and stored at −80 °C. Because we observed the loss of APX activity during storage, in the case of APX assessment, the pellets were not frozen but immediately used for enzyme extraction and determination of its activity.

For the extraction of TBARS, low-molecular-weight antioxidants, and enzymes, algal cells were suspended in 0.6 or 0.65 mL (the latter in the case of proline determination) of the appropriate solution/buffer (see below) and then lysed by sonication (Vibra-Cell VC 505, Sonics, NT, USA) at 35% amplitude, with 30 s total time of ultrasound emission in 10 s on/10 s off cycles. Sonicated samples were centrifuged: 5 min in the case of low-molecular-weight antioxidants, 15 min in the case of enzymes, and 30 min in the case of TBARS, at 11,000× *g*, 4 °C). The supernatants were then used to determine compound content or enzyme activity. If there was a need for short-term storage of extracts, they were kept on ice and protected from light.

For the assessment of O_2_^−•^ generation, equal volumes of each replicate were mixed together per each series, and then, volumes of algal culture containing 14.3 × 10^6^ cells were taken per each series.

### 4.4. TBARS Measurements

Lipid peroxidation was measured in terms of TBARS, according to a slightly modified protocol described by Elbaz et al. [48]. Algal pellets were extracted using 5% trichloroacetic acid (TCA) as described above, and then, 0.5 mL of extract was mixed with 0.5 mL of 0.5% thiobarbituric acid (TBA) in 20% TCA. The reaction mixture was incubated for 30 min at 95 °C, then cooled on ice and centrifuged (5 min, 9000× *g*). The absorbance of the supernatant was measured at λ = 532 nm and 600 nm. The value of nonspecific absorbance at 600 nm was subtracted, and the amount of TBARS was calculated using the extinction coefficient of 155 mM^−1^ cm^−1^.

### 4.5. The Assessment of Low-Molecular-Weight Antioxidants

For the determination of Asc, DHA, and soluble thiols, algal pellets were extracted as previously described with 5% TCA. For proline extraction, 3% sulphosalicylic acid was used. The ascorbate and DHA content was determined according to Kampfenkel et al. [49]. Total soluble thiols were assessed using Ellman’s reagent [50]. Free Pro was determined according to Bates et al. [51]. Methods were optimised for experiments with *C. reinhardtii* [44].

For the determination of proline, the following procedure was applied: 0.6 mL of extract was mixed with 0.6 mL of freshly prepared ninhydrin reagent (10.8 mL of glacial acetic acid, 7.2 mL of 6 M H_3_PO_4_, and 0.45 g of ninhydrin), and then, 0.6 mL of glacial acetic acid was added. The mixtures were incubated in glass vials at 100 °C for 1 h, then cooled on ice for 5 min and extracted with toluene (1.2 mL of toluene per sample, followed by vortexing for 15–20 s). The absorbance of the toluene phases was measured at λ = 520 nm.

For the determination of thiols, 150 µL of extract was mixed with 1 mL of 0.3 M TRIS-HCl buffer pH 8.2, and then, 115 µL of freshly prepared 10 mM 5,5′-dithiobis-(2-nitrobenzoic) acid (in methanol) was added. The mixtures obtained were incubated on a shaker for 20 min at room temperature, and then, the absorbance was measured at λ = 412 nm.

For the determination of total ascorbate (Asc + DHA), the following procedure was applied: 140 μL of extract + 150 μL 150 mM potassium phosphate buffer pH 7.4 + 10 μL 3 M KOH + 50 μL H_2_O + 50 μL 10 mM dithiotreitol (DTT) dissolved in 150 mM potassium phosphate buffer pH 7.4 (followed by 15 min of incubation at room temperature) + 50 μL 0.5% N-ethylmaleimide (NEM) dissolved in water (followed by vigorous mixing for 1 min) + 300 μL 10% TCA + 300 μL 44% H_3_PO_4_ + 300 μL 4% 2,2′-bipirydyl dissolved in 70% ethanol (*v*/*v*) + 150 μL 3% FeCl_3_∙6H_2_O *w*/*v* (followed by vigorous mixing for 30 s). The reaction mixtures obtained were incubated for 1 h at 40 °C, and then, the absorbance was measured at λ = 525 nm. For the determination of Asc alone, the steps with the addition of DTT and NEM were omitted, and the same volume of H_2_O was added instead.

The calibration curves were prepared accordingly, using Asc, Cys, and Pro standards bought from Sigma-Aldrich. Antioxidant content was normalised to known total protein content in the samples collected from 30 mL of cultures (see below).

### 4.6. The Assessment of Antioxidant Enzymes Activity

For the determination of SOD and CAT, the algal pellets were extracted as described above using freshly prepared 50 mM potassium phosphate buffer pH 7.0 containing 1 mM phenylmethylsulphonyl fluoride, 1% polyvinylpyrrolidone, 0.5 mM Na_2_EDTA, and Triton X (0.25 mL of detergent for 100 mL of buffer). The buffer used for the extraction of APX also contained 1 mM ascorbic acid. SOD activity was determined by measuring the inhibition of photochemical reduction of nitroblue tetrazolium (NBT) [52], CAT activity according to method described by Aebi [53], and APX activity according to Nakano and Asada [54]. Methods were optimised for experiments with *C. reinhardtii* [33,44].

For the determination of CAT, 100 µL of extract was added to 3 mL of 50 mM potassium phosphate buffer pH 7.0, and then, 34 µL of 3% H_2_O_2_ was added, and its consumption was measured by absorbance detection at λ = 240 nm. For each extract, the measurement was carried out twice. CAT activity was expressed in units where 1 unit of CAT converts 1 µmole of H_2_O_2_ per min (H_2_O_2_ molar extinction coefficient = 0.036 mM^−1^ cm^−1^).

For the determination of APX, 250 µL of extract was added to 3 mL of 50 mM potassium phosphate buffer pH 7.0 containing 0.5 mM ascorbic acid and 0.25 mM Na_2_EDTA, and then, 34 µL of 3% H_2_O_2_ was added, and Asc consumption was measured by absorbance detection at λ = 290 nm. For each extract, the measurement was carried out twice. The series with boiled extract was also measured to assess the rate of decrease in Asc due to direct reaction with H_2_O_2_. APX activity was expressed in units where 1 unit of APX converts 1 µmole of Asc per min (Asc molar extinction coefficient = 2.8 mM^−1^ cm^−1^).

For SOD determination, the reaction mixtures were prepared in 48-well plates by pipetting the following components into each well: 750 µL of freshly prepared reaction buffer (50 mM potassium phosphate buffer pH 7.8 containing 10 mM methionine and 75 µM NBT), 50 µL of extract (or extraction buffer in the case of enzyme-free series), and 7 µL 100 µM riboflavin (in water). The plate was then exposed to white light at the intensity of 870 μmol photons m^−2^ s^−1^. Absorbance at λ = 560 nm was measured at t = 0, 2, 4, 6, 8, 10, and 12 min using a microplate reader (Infinite 200 Pro, Tecan Life Sciences, Männedorf, Switzerland). For each extract, the measurement was carried out twice. The activity of SOD was expressed in units where 1 unit of SOD is a quantity of enzyme inhibiting the reduction of NBT by 50% compared to the enzyme-free reaction mixture.

The total soluble protein content was determined in the algal extracts using Bradford reagent, according to the protocol provided by the supplier (Sigma-Aldrich, Taufkirchen, Germany). The enzyme activity was then normalised to the total protein content.

### 4.7. Semi-Quantitative Analysis of Cumulative Superoxide Production

Cumulative O_2_^−•^ production in *C. reinhardtii* cells was analysed using in vivo NBT staining [55] with modifications [44]. Cell suspensions were centrifuged (600× *g*, 8 min, 10 °C). The pellet was then re-suspended in 2 mL of the growth medium with basal Cu concentration containing 1 mM NBT. Then, 200 µL of each suspension was spotted on Whatman filter paper (Whatman 3MM Chr) in nine replications. The paper was wrapped in transparent food wrap to protect it from water evaporation. The algae were then incubated for 20 min at 21 °C under illumination 35–40 μmol photons m^−2^ s^−1^. After incubation, photosynthetic pigments were removed by acetone extraction. The dried papers on which the blue precipitate of formazan could be seen were scanned using Epson Perfection V550 Photo. Semi-quantitative densitometric analysis was performed using Fiji ImageJ software ver. 2.9.0.

### 4.8. The Assessment of Chromium Tolerance

The set of modified SG media with basal concentration of Cu was prepared: control medium and media containing 10, 20, 40, 80, 120, 160, and 240 μM K_2_Cr_2_O_7_. Cell suspensions of the N1 and Cu2 strains were centrifuged (600× *g*, 8 min, 10 °C). The pellets were re-suspended in fresh SG medium with basal concentration of Cu. The algae were then inoculated into control and Cr-containing media to provide an initial Chl *a* + *b* concentration of 0.05 μg/mL. Cell suspensions were then portioned into 48-well sterile plates with 1 mL per well and six replicates for each series. The algae were then grown for 1 week at 21 °C under illumination 35–40 μmol photons m^−2^ s^−1^.

Starting from day 3, the Chl fluorescence was measured each day using a PAM-101 chlorophyll fluorometer (Walz, Effeltrich, Germany) with a red (600–680 nm) saturating light source (5000 µmol photons m^−2^ s^−1^). The output of the light guide was focused on the bottom of the plate; a cover with the hole adjusted to the well diameter was placed between the end of the optical fibre and the bottom side of the plate.

### 4.9. Statistical Analysis

Statistical analysis of the data presented in Figure 1 and Figure 2 and the Table 1 and Table 2 was carried out using STATISTICA 13.3. The following analyses were carried out: two-way ANOVA, post hoc Tukey’s test to compare means, and correlation analysis. Statistical analysis of the data presented in Figure 3 and Table 3 was carried out using R-4.3.2 [56], using the lm(), aov(), and summary() functions, to obtain parameters of linear model.

## 5. Conclusions

The Cu-tolerant strain Cu2 showed less pronounced lipid peroxidation and increased APX activity compared to the N1 strain. However, the amount of hydrophilic low-molecular-weight antioxidants measured was similar in both strains. The activities of SOD and CAT were lower in the Cu2 strain when compared to the N1 strain. Exposure to excessive Cu led to a similar increase in Pro content in both strains and a decrease in Asc and thiols, which was more pronounced in the N1 strain. The Cu2 strain displayed diminished tolerance to another redox-active heavy metal, Cr, compared to the N1 strain. This lets us conclude that other mechanisms probably connected to Cu transport, partitioning, and chelation are more important for Cu tolerance in the Cu2 strain than the antioxidant enzymes and compounds measured. The results obtained show that algal strains tolerant to certain heavy metal ions may not be useful for the treatment of wastewater containing a mixture of ions of various heavy metals.

## Figures and Tables

**Figure 1 plants-13-00999-f001:**
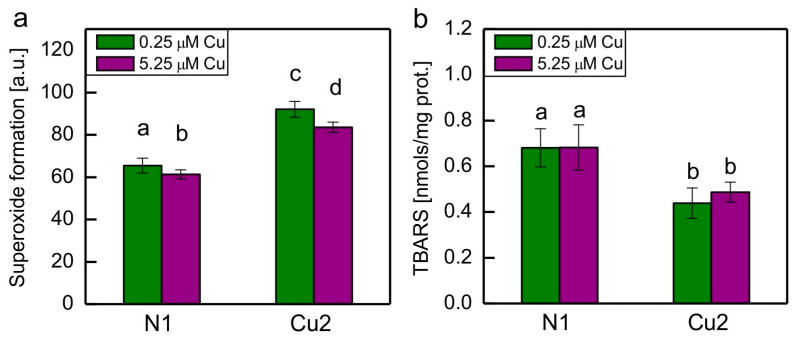
Superoxide formation (**a**) and thiobarbituric acid-reactive substances (TBARS) content (**b**) in N1 and Cu2 *C. reinhardtii* strains grown for 7 days in the medium containing either basal (0.25 μM) or increased (5.25 μM) concentration of CuSO_4_. Data are means ± SD (*n* = 9 for O_2_^−•^ measurements, *n* = 4 for TBARS determination). Different letters denote means that differ from each other with statistical significance *p* < 0.05 (post hoc Tukey’s test).

**Figure 2 plants-13-00999-f002:**
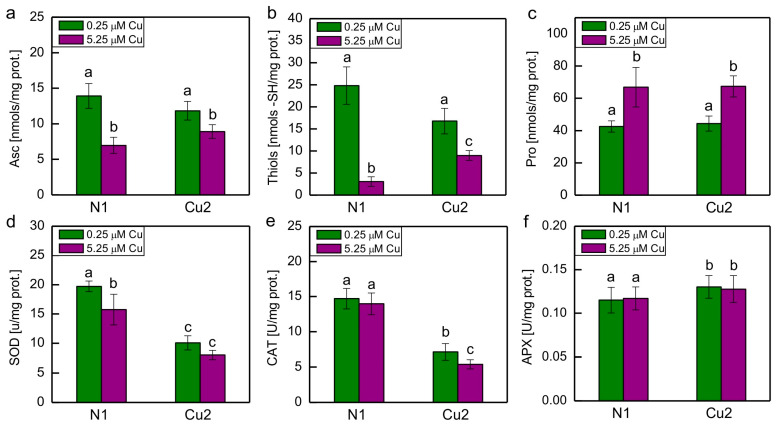
The content of ascorbate (**a**), total soluble thiols (**b**), and proline (**c**) and the activity of superoxide dismutase (**d**), catalase (**e**), and ascorbate peroxidase (**f**) in N1 and Cu2 *C. reinhardtii* strains grown for 7 days in the medium containing either basal (0.25 μM) or increased (5.25 μM) concentration of CuSO_4_. Data are means ± SD (*n* = 4 for low-molecular-weight antioxidants, *n* = 8 for antioxidant enzymes). Different letters denote means that differ from each other with statistical significance *p* < 0.05 (post hoc Tukey’s test). APX, ascorbate peroxidase; Asc, ascorbate in its reduced form; CAT, catalase; Pro, proline; SOD, superoxide dismutase.

**Figure 3 plants-13-00999-f003:**
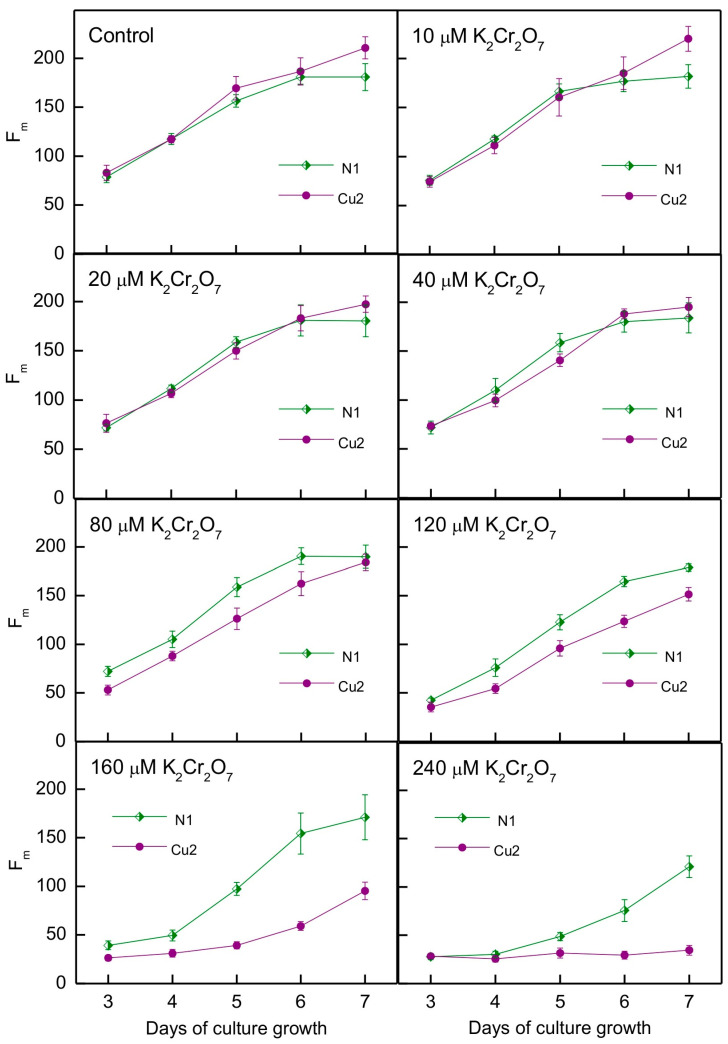
Growth of N1 and Cu2 *C. reinhardtii* strains in the media containing different concentrations of K_2_Cr_2_O_7_, measured as maximal chlorophyll fluorescence. Data are means ± SD (*n* = 6).

**Figure 4 plants-13-00999-f004:**
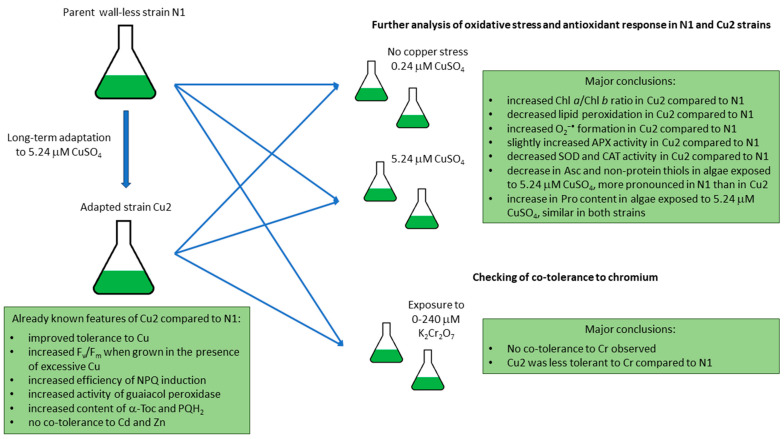
Summary of the present experiment. Already known features of Cu2 strain have been described in [5,7]. APX, ascorbate peroxidase; Asc, ascorbate; CAT, catalase; Chl, chlorophyll; F_v_/F_m_, maximum quantum yield of photosystem II; NPQ, nonphotochemical quenching of chlorophyll fluorescence; PQH_2_, plastoquinol; Pro, proline; SOD, superoxide dismutase; α-Toc, α-tocopherol.

**Table 1 plants-13-00999-t001:** Photosynthetic pigments ratios in N1 and Cu2 *C. reinhardtii* strains grown for 7 days in the medium containing either basal (0.25 μM) or increased (5.25 μM) concentration of CuSO_4_. Data are means ± SD (*n* = 4). Different letters denote means that differ from each other with statistical significance *p* < 0.05 (post hoc Tukey’s test). Car, total carotenoids; Chl *a*, chlorophyll *a*; Chl *b*, chlorophyll *b*.

Strain	Chl *a*/Chl *b*		Car/(Chl *a* + Chl *b*)	
	Basal Cu	Increased Cu	Basal Cu	Increased Cu
N1	1.96 ± 0.04 ^a^	1.99 ± 0.08 ^a^	0.22 ± 0.01 ^a^	0.22 ± 0.01 ^a^
Cu2	2.14 ± 0.05 ^b^	2.09 ± 0.06 ^b^	0.23 ± 0.01 ^a^	0.23 ± 0.01 ^a^

**Table 2 plants-13-00999-t002:** Matrix of correlations between antioxidant content, activity of antioxidant enzymes, lipid peroxidation, copper concentration in the medium, and copper adaptation in the strains. Results that were statistically significant (*p* < 0.05) are marked with colour: red for positive correlation and blue for negative correlation. APX, ascorbate peroxidase; Asc, ascorbate in its reduced form; Car_n_, carotenoids normalised on chlorophyll content; CAT, catalase; Cu_adapt_, adaptation for copper; Cu_conc_, copper concentration in the medium; Pro, proline; SOD, superoxide dismutase; TBARS, thiobarbituric acid reactive substances; Thiols, total non-protein thiols. Results for superoxide were excluded because of different sample collection pattern (see Section 4.3).

Asc	Thiols	Pro	CAT	APX	SOD	TBARS	Car_n_	Cu_conc_	Cu_adapt_	
1	0.959	−0.761	0.348	−0.221	0.448	0.272	0.026	−0.846	−0.201	Asc
	1	−0.778	0.227	−0.236	0.364	0.091	0.008	−0.890	−0.112	Thiols
		1	−0.338	0.309	−0.400	−0.120	0.071	0.854	0.238	Pro
			1	−0.351	0.955	0.863	−0.546	−0.322	−0.952	CAT
				1	−0.233	−0.351	0.572	0.173	0.437	APX
					1	0.778	−0.443	−0.441	−0.920	SOD
						1	−0.577	−0.067	−0.824	TBARS
							1	−0.055	0.592	Car_n_
								1	0.169	Cu_conc_
									1	Cu_adapt_

**Table 3 plants-13-00999-t003:** Statistical analysis of *C. reinhardtii* growth (log F_M_) in relation to experimental population, K_2_Cr_2_O_7_ concentration, and time.

	F	*p*
Strain	82.04	<2 × 10^−16^
Cr concentration	2118.536	<2 × 10^−16^
Day	2786.637	<2 × 10^−16^
Strain × Cr concentration	130.374	<2 × 10^−16^
Strain × day	7.889	0.00518
Cr concentration × day	80.66	<2 × 10^−16^
Strain × Cr concentration × day	95.519	<2 × 10^−16^

## Data Availability

The original contributions presented in the study are included in the article material; further inquiries can be directed to the corresponding author.

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
