# Peer review of "Unravelling the Mechanisms of Heavy Metal Tolerance: Enhancement in Hydrophilic Antioxidants and Major Antioxidant Enzymes Is Not Crucial for Long-Term Adaptation to Copper in Chlamydomonas reinhardtii"

_plants, 2024, doi:10.3390/plants13070999_

Round 1

Reviewer 1 Report

Comments and Suggestions for Authors

 This work shows that several antioxidant systems of Chlamydomonas reinhardtii are not involved in the cellular response to stress caused by excess copper. In this regard, it would be useful in the discussion section to analyze possible other components that ensure tolerance to copper of the mutant used in the work, taking for comparison data on plants and yeast, for example, presented in reviews:

Ejaz U, Khan SM, Khalid N, Ahmad Z, Jehangir S, Fatima Rizvi Z, Lho LH, Han H, Raposo A. Detoxifying the heavy metals: a multipronged study of tolerance strategies against heavy metals toxicity in plants. Front Plant Sci. 2023 May 12;14:1154571. doi: 10.3389/fpls.2023.1154571. PMID: 37251771; PMCID: PMC10215007.

Robinson JR, Isikhuemhen OS, Anike FN. Fungal-Metal Interactions: A Review of Toxicity and Homeostasis. J Fungi (Basel). 2021 Mar 18;7(3):225. doi: 10.3390/jof7030225. PMID: 33803838; PMCID: PMC8003315.

Author Response

Dear Reviewer,

Thank you very much for  the review and the remark. Other mechanisms of heavy metal tolerance have been included in the discussion. The added text was highlighted. The suggested papers were cited, together with few more.

Yours sincerely,

Beatrycze Nowicka

Reviewer 2 Report

Comments and Suggestions for Authors

The submitted manuscript to PLANTS-MDPI entitled “Antioxidant response and tolerance to chromium-induced stress in copper-tolerant Chlamydomonas reinhardtii strain” is interesting to investigate. BUT, following are the comments that need to be addressed:

The title is not so attractive. It should be modified.

If the authors selected non-tolerant species too, then why did they mention the tolerant one in title which makes it vague.

Line 17: It was expected the tolerant species will have a lower MDA level, but what was the reason behind it in your case?

On what basis did the authors choose to study these two concentrations of Cu?

In my opinion, authors have presented their results well, but it would be very nice if authors could draw a mechanistic diagram to explain their study in the discussion section.

In addition, I would suggest doing the PCA and correlation analysis.

Author Response

Dear Reviewer,

Thank you very much for  the review and the remarks. The text was changed accordingly and the modifications were highlighted.

>The title is not so attractive. It should be modified.

>If the authors selected non-tolerant species too, then why did they mention the tolerant one in title which makes it vague.

The title was changed. The non-tolerant parent strain was used as a control, therefore it was not mentioned (we didn’t want the title to be too long). Now the title is a little different.

>Line 17: It was expected the tolerant species will have a lower MDA level, but what was the reason behind it in your case?

We made it more precise in the abstract explaining that alpha-tocopherol and plastoquinol, which content is increased in Cu2 strain, are lipophilic antioxidants. We also extended the explanation in the discussion section (lines 276-277).

>On what basis did the authors choose to study these two concentrations of Cu?

We added explanation in the Material and Methods sections 4.1 and 4.2.

> In my opinion, authors have presented their results well, but it would be very nice if authors could draw a mechanistic diagram to explain their study in the discussion section.

The figure was added to the discussion section as Figure 4.

>In addition, I would suggest doing the PCA and correlation analysis.

The correlation analysis has been carried out and the results were included in subsection 2.1.

Yours faithfully, 

Beatrycze Nowicka